# Food frequency questionnaire for foods high in sodium: Validation with the triads method

**Diana S. Souza** \*, Bianca I. Santos, Brenda M. Costa, Dalila M. Santos, Laryssa G. S. Aragão, Liliane V. Pires, Diva A. S. Vieira, Analícia R. S. Freire, Kiriaque B. F. Barbosa

Department of Nutrition, Federal University of Sergipe, São Cristóvão, Brazil

\* idiana-ss@hotmail.com

**Data Availability Statement:** All relevant data are within the paper and its Supporting Information files.

## Abstract

This study aimed to validate a food frequency questionnaire for foods high in sodium (FFQ-FHS) in a population aged ≥18 years and to test its reproducibility. This cross-sectional study included 50 individuals (≥18 years) of both sexes. In addition to the FFQ-FHS, four 24-h dietary recalls (24hRs) were conducted and a socioeconomic and lifestyle questionnaire was administered. Two 24-h urinary excretions were collected for sodium analysis, and anthropometry was performed. For validation, the triad method was applied using the validity coefficient (ρ). For reproducibility, the intraclass correlation coefficient (ICC), 95% confidence interval, kappa coefficient, and Bland–Altman plots were used to check for agreement. The Kolmogorov-Smirnov test was used to verify the data distribution. The validity coefficients for daily energy-adjusted sodium intake were high for the 24hR ($\rho_{RAI} = 0.85$) and weak for the FFQ-FHS ($FFQ_{AI} = 0.26$) and biomarker ($\rho_{BAI} = 0.20$). The ICC values were 0.68 for unadjusted sodium and 0.54 for energy-adjusted sodium intake. The weighed Kappa scores were 0.49 (p<0.01) and 0.260 (p = 0.02) for unadjusted and adjusted sodium intake, respectively. Although the FFQ-FHS is reproducible, it is not valid for the assessment of sodium intake and cannot be the sole instrument used for this purpose.

## Introduction

Several studies [1–3] have developed food frequency questionnaires (FFQ) to estimate the sodium intake in different populations; however, validation is necessary. For this purpose, the FFQ should be compared with other methods of food intake assessment, such as food record (FR) and/or 24-hour dietary recall (24hR), which, although they are also prone to measurement errors, are considered reference methods. Biomarkers (Bs) are commonly used in validation studies because they have distinct and independent approaches from other methods that assess dietary intake. However, B have limitations in reflecting the intake of a particular nutrient over a short period. In addition, metabolic reactions, absorption, and excretion affect the B levels [4–7].

When nutrient intake is assessed using two dietary assessment tools and biological markers, the triad method is recommended because it estimates the validity coefficient between the unknown actual intake (AI) and the intake estimated using the FFQ, 24hR or FR, and B, allowing a triangular comparison between the different methods [4, 5, 8, 9]. The triad method is effective in comparing intake estimations using the instrument to be validated, the FFQ, the

**Funding:** This study was partially financed by the following: 1. Scholarship awarded by the Fundação de Apoio à Pesquisa e a Inovação Tecnológica do Estado de Sergipe (Fapitec/SE/Brasil) [Research and Technological Innovation Support Foundation in the State of Sergipe] https://fapitec.se.gov.br/ 2. Financial resources granted by the Coordenação de Aperfeiçoamento de Pessoal de Nível Superior - Brasil (CAPES/Brasil) [Coordination for the Improvement of Higher Education Personnel], financing code 001, PROMOB, process no. 88881.157882/2017-01 https://www.gov.br/capes/pt-br The funders had no role in the study design, data collection and analysis, decision to publish, or manuscript preparation. No additional external funding was received for the study.

**Competing interests:** The authors have declared that no competing interests exist.

reference method (24hR or FR), and B. The proposed method assumes the existence of independent random errors in the three instruments and considers that each error is linearly related to the AI [5].

A systematic review investigated whether FFQs are reliable and valid for measuring the habitual sodium intake and concluded that the available instruments were inadequate because of the low agreement between intake measured by FFQ and 24-hour urinary excretion [10].

In Brazil, independent of the geographical region, the sodium intake of the population is high [11, 12] and corresponds to approximately twice the intake recommended by the World Health Organization (WHO) (2 g/day) [13]. Data from the 2008 to 2009 Household Budget Survey show that the contribution of processed and ultra-processed foods to the average sodium intake increased from that observed in the first edition of the survey (2002–2003) [11]. Despite this, studies seeking the validation of a tool to investigate sodium intake in the Brazilian population, especially in the northeast region, remain scarce. Thus, this study aimed to use the triad method to validate the food frequency questionnaire for foods high in sodium (FFQ-FHS) for Brazilian adults and determine its reproducibility.

## Materials and methods

### Study design and participants

This was a cross-sectional study using a convenience sample. This study enrolled 50 adult Brazilian men and women. This sample size was considered the minimum sample size suggested for validation studies, as recommended by Cade et al. [14] and McLean et al. [10]. The study was approved by the Research Ethics Committee of the Federal University of Sergipe (process no. 1.875.674, CAAE no. 62502216.0.0000.5546).

Participants were recruited through social media and traditional methods (posters and e-mails) in 2018. Fifty male and female individuals aged between 18 and 60 years and belonged to the academic community of a Federal Public University (undergraduate students and staff personnel) were selected. We excluded pregnant and lactating women, vegetarians, individuals clinically diagnosed with non communicable chronic diseases or following special diets, individuals with identifiable cognitive deficits, and nutrition professionals were not included.

All four assistants were nutrition graduate students who received training and supervision from the nutritionists. The researcher and assistants conducted body measurements and interviews using a socioeconomic questionnaire. Data collection and interviews were carried out at the university nutrition school clinic. The researcher and assistants conducted four 24hRs and two FFQ-FHS surveys, in addition to the collection of two urine samples over a period of 6 months. Anthropometric measures were used to determine the nutritional status. The socioeconomic and demographic data were also collected. Fig 1 shows a schematic diagram of the study process.

The participants signed a free and informed consent form, allowing voluntary participation, followed by the application of the socioeconomic questionnaire, 24hR and FFQ-FHS, and anthropometric assessment. In the first and fourth interviews, the FFQ-FHS was completed after the 24hR via face-to-face interviews with the support of a food portion photographic book.

### Food frequency questionnaire for foods high in sodium

The FFQ-FHS is a 55-item semiquantitative tool developed by Santos et al. [15] to assess the habitual sodium intake during the preceding 6-month period. Participants in this validation study completed two sets of FFQ-FHS, at the beginning and end of the semester, to assess for reproducibility.

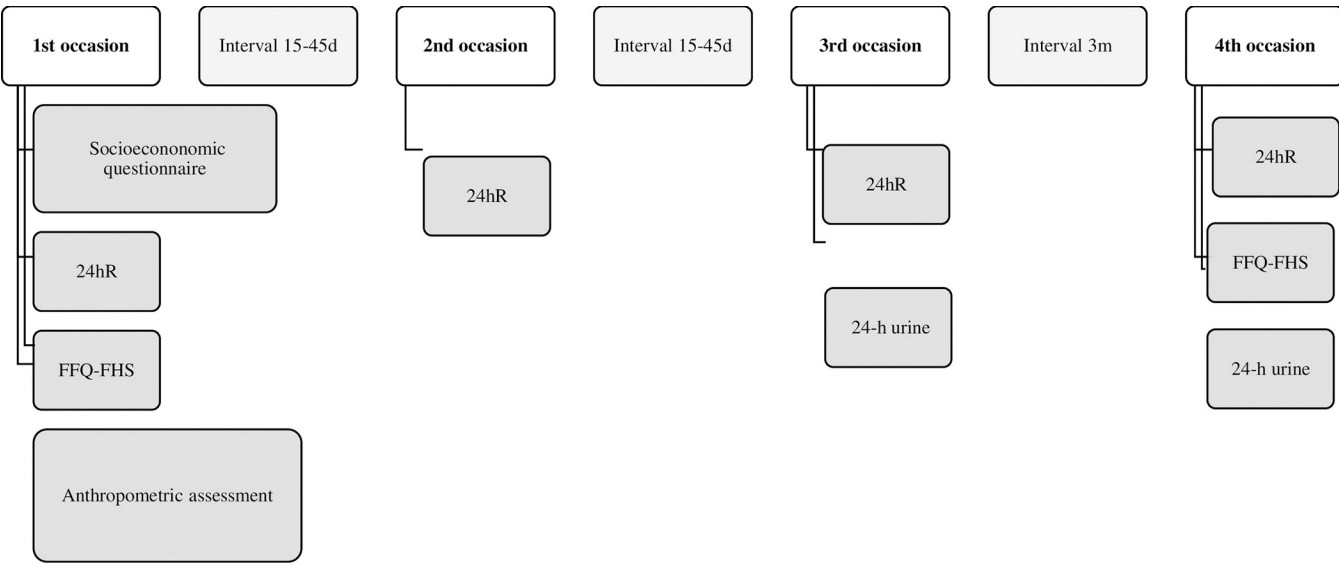

**Fig 1. Data collection process.** Description of data collection process. SQ:sociodemographic questionnaire; 24hR:24-h dietary recall; FFQ-FHS:food frequency questionnaire for foods high in sodium; AA: anthropometric assessment.

The FFQ-FHS contain a list of foods that contain ≥400mg of sodium per 100g or ml, as defined by the National Health Surveillance [16] for high-sodium food. The list of items corresponds to the foods consumed by the local population, selected after applying the 24hR [15]. The foods and typical meals that make up the dietary pattern of the reference population or those that indicated a consumption rate of ≥5% in the northeast region [17] were also included in the FFQ-FHS. The standard portion size of each food or meal was defined according to the Brazilian Institute of Geography and Statistics (*Instituto Brasileiro de Geografia e Estatística*, IBGE) database of reference measurements and portion sizes of foods and dishes consumed in Brazil [18].

The food items in the FFQ-FHS are divided into seven groups (processed meats, canned and preserved foods, dairy products, bakery and pasta, salts and condiments, miscellaneous, and regional foods), four portion options (½ portion, 1 portion, 2 portions, and more portions–the interviewee should specify the amount consumed), and eight intake frequency categories (rarely/never, once a month, 2–3 times a month, once a week, 2–4 times a week, once daily, twice daily, and ≥3 times a day).

## 24-h dietary recall

Each volunteer answered three 24hR questions, with minimum and maximum intervals of 15 and 45 days, respectively, depending on the participants' availability, including one day during the weekend. After 3 months, the fourth 24hR was carried out at the end of the period. The researcher and assistants were trained to conduct the four 24hRs using the multiple pass method, developed by the United States Department of Agriculture [19]. To facilitate the completion of all forms, a photographic manual of food quantification with preset portion sizes was used [20].

## Added salt

In addition to the sodium estimate obtained during the 24hR, the amount of sodium in the added salt was considered. The participants were asked if they had the habit of adding salt to

the dish and salad. If yes, they were asked whether commercial sachets or table salt was used. Considering that each sachet contained 1 g of salt, the units used were counted to estimate the total sodium intake. For those who confirmed the use of table salt, the estimated consumption was 1 g of added salt. This estimate was adopted because the amount of salt added to the table could not be quantified.

### Sodium excretion biomarker

A 24-h urine specimen was collected following the recommendation of McLean et al. [10] to determine the daily excretion of Na (spectrophotometry) and creatinine (*Jaffé* method). Between the third and the fourth 24hR, two 24-hr urine specimens were collected for urinary sodium analysis. The participants received instructions [21] on how to collect and deliver the biological materials. The participants received 2-L containers for 24-h urine collection.

The participant had to discard the first urine of the day and collect all urine from that moment on, including nocturnal and bath urine, until the next day at the same time as that during which the first urine was discarded. The collected urine was stored in a container and kept in a refrigerator during and after collection. The delivered material was identified and sent to the Clinical Laboratory of the Department of Nutrition of the UFS.

Considering the need for a minimum volume of urine to identify the excretion of metabolites, we used urine specimens with less than 400mL [22], or with a creatinine value <0.2 mmol/kg/day [23]. The results were expressed in mg/kg/day.

### Sociodemographic data

A structured questionnaire was used to collect the sociodemographic information (sex, age, household *per capita* income, and educational level). The household *per capita* income was calculated by summing the monetary income reported and dividing by the number of all family members (minimum wage corresponding to USD $ 248.69).

### Anthropometry

The assistants performed anthropometric measurements on the first day of the interviews. Standard and per protocol measurement techniques were used in this study. A digital platform scale (LIDER®, São Paulo, Brazil) was used to determine the body weight with a capacity of 200kg and a precision of 100g. Stature was measured using a portable 2-m stadiometer with a precision of 1 millimeter (Alturaexata®, Minas Gerais, Brazil). Waist circumference was measured in centimeters using inelastic anthropometric tape. Nutritional status was classified according to the body mass index based on the cut-off values proposed by the WHO [24].

### Food composition analysis

The food nutrient composition was analyzed using NutWin® software (version 1.5.2.51). All items listed in the applied 24hR and FFQ-FHS were standardized and typed individually in duplicate. The IBGE [25] nutrient composition table and product labels (mean values of the three brands) were used.

### Statistical analysis

The Kolmogorov-Smirnov test was used to determine normality of data distribution. Descriptive statistics was conducted for calculation of the measures of central tendency and dispersion or absolute or relative frequencies. Spearman's correlation coefficients were used to determine agreement between the FFQ-FHS, 24hR, and B.

The FFQ-FHS validity was analyzed using the triad method, comparing the results of the FFQ-FHS survey, 24hR, and B assessment with the estimated AI. The validity coefficient was determined, which can vary from 0 to 1 and must not be negative. The correlations were considered low (r<0.3) or high (r>0.7) [5]. To check for agreement between FFQ-FHS, 24hR, and B, the weighted Kappa value was calculated using the daily energy-adjusted sodium intake categorized into tertiles.

For reproducibility, the correlation between the two FFQ-FHS surveys was assessed using the intraclass correlation coefficient (ICC) and a confidence interval of 95% (95% CI). The weighted kappa value was used to check the agreement of sodium classifications categorized into tertiles.

Bland–Altman [26] plots graphically demonstrate the level of agreement between the FFQ, 24hR, and B, and the reproducibility of the FFQ-FHS. The centerline indicates the bias. It is desirable that this line be positioned close to zero, which means that there is a small difference between the means of both instruments. The upper and lower limits represent the 95% CI [27]. Good agreement is also shown by a greater number of points concentrated around the centerline. In the plots, the disagreement is expressed by a great dispersion of points around the centerline; additionally, a proportional bias exists because the differences diminish as the means increase [26, 27].

The 24hR data were attenuated to correct for intra-individual variability using the Multiple Source Method [28]. The method estimates the individual usual intake based on the results of repeated measurements over a long period. The program estimates the probability of nutrient intake from the data provided and then estimates the usual intake of nutrients. The values obtained were multiplied with each other to obtain an estimate of the individual usual daily intake after removing the intra-individual variability. The significance level was set at p<0.05.

## Results

The sample included 60% of men, 84% of individuals with secondary education, and 62% of individuals whose household *per capita* income was lower than one minimum wage. The participants' average age was 22.5 years. With regard to nutritional status, 60% of the patients had normal weight. The prevalence of overweight and obesity was 30% (Table 1).

A total of 47 and 36 specimens from the first and second urine collections were adequate according to the urine volume [22] and creatinine value [23].

The mean sodium intake and urinary excretion values are presented in Table 2. The estimated values were higher after administering the first set of FFQ-FHS compared with that after the second set. In both cases, the sodium intake was above the recommended level. The level of urinary sodium excretion exceeded the average dietary intake estimated by both the 24hR and FFQ-FHS survey. The mean sodium estimated in the four 24hRs was also higher than that in the two FFQ-FHS surveys.

The Spearman's correlations and validity coefficients are shown in Table 3. A statistical significance was only observed in the unadjusted sodium correlation between the FFQ and 24hR (r = 0.386). The validity coefficient was high for the 24hR (r>0.7). The validity coefficient for unadjusted sodium considering the average urinary excretion could not be verified because of the negative *r*RB. The analysis of the correlation coefficients indicates that the particular properties of sodium make it difficult to estimate the consumption.

Classification by tertiles of energy-adjusted sodium intake using the three methods indicated a disagreement. The best agreement was observed between 24hR and B assessment (p = 0.271) (Table 4). A tertile analysis was performed to classify the individuals' sodium consumption. The percentages indicated in the same tertile reveal the agreement of classification of the methods, whereas the opposite and adjacent tertiles indicate the degree of disagreement.

**Table 1. Socioeconomic and demographic characteristics and nutritional status of the academic community.** Aracaju, SE, 2018 (n = 50).

| Characteristic | $\bar{X} \pm$ SD or n(%) or |
|---|---|
| **Age (years)** | 22.5±3.5 |
| **Sex** | |
| Male | 30 (60) |
| Female | 20 (40) |
| **Educational level** | |
| Secondary education | 42 (84) |
| Higher education | 8 (16) |
| ***Per capita* income** | |
| ≤1 MW | 31 (62) |
| 2–3 MW | 15 (30) |
| 4–6 MW | 3 (6) |
| ≥9 MW | 1 (2) |
| **Body mass index (kg/m$^2$)** | |
| Thin | 5 (10) |
| Normal | 30 (60) |
| Overweight | 10 (20) |
| Obese | 5 (10) |

Data are shown as mean ± standard deviation [(±DP)] or absolute and relative frequency (n [%]).

Household *per capita* income was calculated by summing the monetary income reported and dividing it by the number of all family members (minimum wage corresponding to USD $ 248.69).

Bland–Altman plots (Fig 2) were constructed to assess the agreement between the FFQ, 24hR, and B, and the reproducibility of the FFQ-FHS. A disagreement was observed between the instruments evaluated. The reproducibility graphs indicated a moderate agreement between the first FFQ-FHS survey and second FFQ-FHS survey as most of the points were placed around the centerline.

**Table 2. Estimated values of unadjusted sodium intake and energy-adjusted sodium intake by the 24hR, FFQ-FHS survey, and biomarker assessment in individuals of the academic community.** Aracaju, SE, 2018 (n = 50).

| Instrument | Sodium (mg) |
|---|---|
| | X±SD |
| **24hR** | 3421.8±909.4* |
| | 3422.8±500.3† |
| **FFQ 1** | 3749.2±2097.4* |
| | 3356.5±333.3† |
| **FFQ 2** | 3100.8±1422.4* |
| | 2838.6±270.9† |
| **Urinary excretion** | (1$^{st}$ collection, n = 47) 4844±1650.1 |
| | (2$^{nd}$ collection n = 36) 4812.6±1778.7 |

24hR = 24-h dietary recall, FFQ = food frequency questionnaire, X = mean, SD = standard deviation

*Sodium unadjusted intake

†Sodium adjusted for daily energy intake

**Table 3. Spearman's correlation and validity coefficients for unadjusted and energy-adjusted daily sodium intake in adults in the academic community.** Aracaju, SE, 2018 (n = 50).

| | Correlation coefficient | | | Validity coefficient | |
|---|---|---|---|---|---|
| | Unadjusted (p) | Energy Adjusted (p)* | | | Energy Adjusted* |
| **rFFQR** | 0.386 (<0.01) | 0.129 (0.387) | **ρFFQAI** | | 0.26 |
| **rRB†** | −0.026 (0.882) | 0.172 (0.316) | **ρRAI†** | | 0.85 |
| **rFFQB†** | 0.014 (0.936) | 0.052 (0.763) | **ρBAI†** | | 0.20 |

rFFQR, correlation between the food frequency questionnaire survey and the 24-h dietary recall

rRB, correlation between the 24-h dietary recall and biomarker assessment

rFFQB, correlation between the food frequency questionnaire survey and biomarker assessment

ρFFQAI, validity coefficient of the food frequency questionnaire survey

ρRAI, validity coefficient of the 24-h dietary recall

ρBAI, biomarker validity coefficient

$\rho FFQAI = \sqrt{([rFFQR \times rFFQB])/rRB}$

$\rho RAI = \sqrt{([rFFQR \times rRB])/rFFQB}$

$\rho BAI = \sqrt{([rRB \times rFFQB])/rFFQR}$

*Daily energy-adjusted sodium intake

†n,36 using the average urinary sodium excretion

The correlation and reproducibility coefficients of FFQ-FHS were moderate. The classification by tertiles of sodium intake and energy-adjusted sodium as estimated by the FFQ-FHS survey showed the exact agreement rates between 50% and 66% (Table 5).

## Discussion

The correlations and validity coefficients found in this study were ranked as low to moderate. The correlation of unadjusted sodium between the FFQ survey and 24hR was weak, but was significant (r = 0.386; p<0.01), which eventually lost significance after adjustment for energy intake. The minimum correlation was 0.4 for the relative validity of an FFQ [29]. FFQ validation studies using FR or 24hR as a reference method also found non-significant associations and, sometimes, negative associations. Sodium correlations varied from 0.07 to 0.43 [1, 30–32].

The validity coefficients for daily energy-adjusted intake were considered high for the 24hR (ρRAI = 0.85) and weak for the FFQ-FHS survey (ρFFQAI = 0.26) and B assessment (ρBAI = 0.20). The findings of this study were similar to those of Pereira et al. [1], in which the B assessment showed the weakest validity coefficient (ρBAI = 0.21), while the 24hR showed the highest validity coefficient (ρRAI = 0.56). The validity coefficients based on unadjusted sodium levels were not verifiable owing to the negative correlation. Negative correlations have been reported in previous studies [9, 33].

**Table 4. Weighed kappa agreement ranked by tertiles of energy-adjusted daily sodium intake in adults in the academic community.** Aracaju, SE, 2018 (n = 50).

| Method | Same tertile (%) | Adjacent tertile (%) | Opposite tertile (%) | Weighed Kappa | p |
|---|---|---|---|---|---|
| FFQx24hR | 32 | 44 | 24 | −0.041 | 0.772 |
| FFQxB§ | 27.8 | 47.2 | 25 | −0.82 | 0.624 |
| 24hRxB§ | 52.8 | 27.8 | 19.4 | 0.183 | 0.271 |

FFQ:food frequency questionnaire; 24hR: 24-h dietary recall; B: biomarker

§n,36 Rank of average urinary sodium excretion

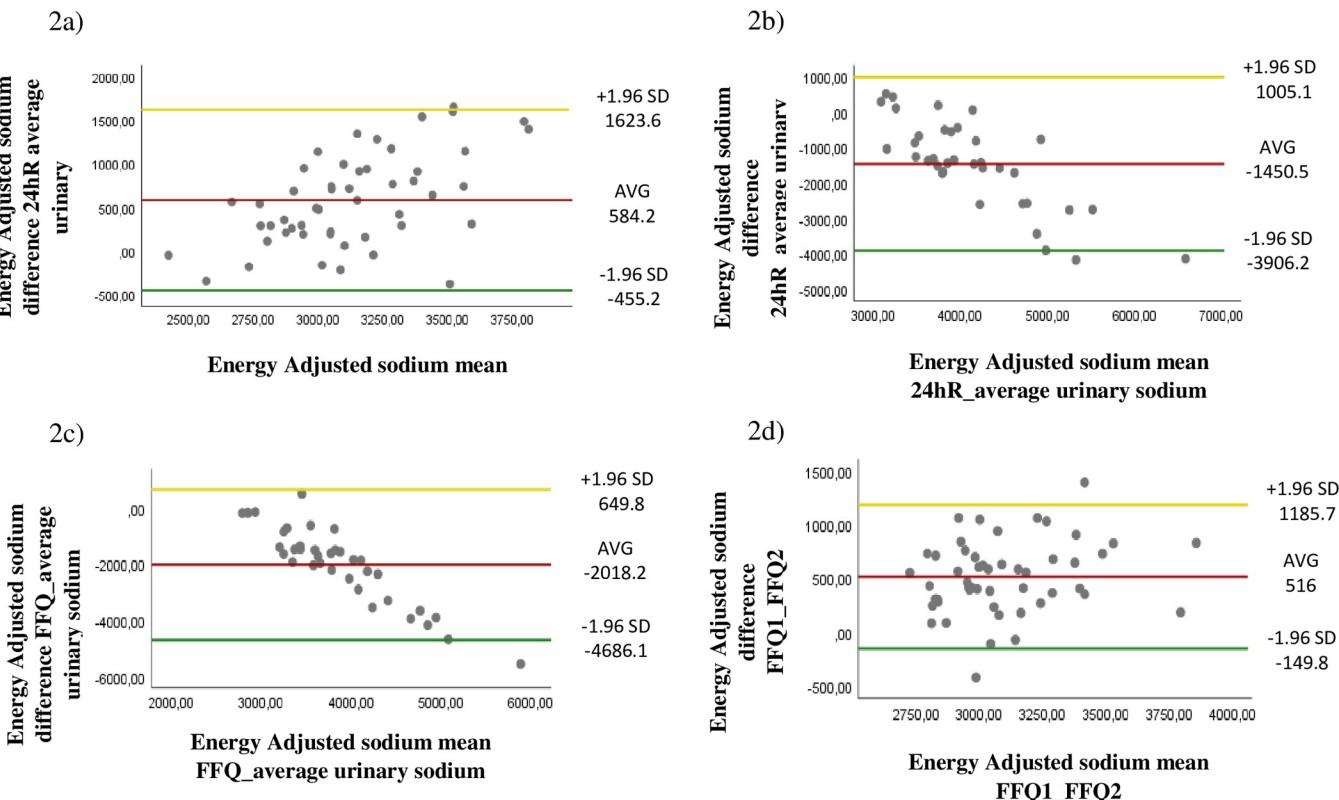

**Fig 2. Bland–Altman plots to assess for agreement of daily energy-adjusted sodium between the FFQ survey, 24hR, and biomarker assessment in individuals from the academic community.** Aracaju, SE, 2018 (n = 50). a) Food frequency questionnaire and 24-h dietary recall; b) 24-h dietary recall and average urinary excretion (n = 36); c) food frequency questionnaire and average urinary excretion (n = 36); d) agreement of the daily energy-adjusted sodium intake between the FFQ-FHS 1 and 2 *Absolute differences between values from the ordinate axes and mean intake of the abscissa axes. SD = standard deviation, AVG = average.

In addition to the calculation of correlation and validity coefficients, the use of different analyses to estimate the validity of an instrument is important. Although the triad method was chosen for this study, other statistical tests were conducted to ensure the greater reliability of results, such as the weighted Kappa agreement and Bland–Altman [26] plots. Furthermore, the recommendations for carrying out an FFQ validation study to assess the sodium intake were followed according to McLean et al.'s study [10].

According to Kaaks [4], the FFQ survey and 24hR exhibit similar errors, which overestimate the correlation and, consequently, the validity coefficient. On the contrary, the correlation between the FFQ survey and B assessment was weak owing to the intra-individual influence; the B reflects the short-term intake, restricted to a short period of time, while the FFQ assesses the long-term intake. Strong correlations (r>0.7) produce high validity coefficients, whereas weak correlations (r<0.3) generate undesirable validity coefficients [4].

**Table 5. Reproducibility of the FFQ-FHS for unadjusted sodium intake and daily energy-adjusted sodium intake and classification by tertiles of sodium intake by individuals from the academic community.** Aracaju, SE, 2018 (n = 50).

| Nutrient | ICC | 95% CI | Same tertile (%) | Adjacent tertile (%) | Opposite tertile (%) | Weighed Kappa | p |
|---|---|---|---|---|---|---|---|
| Unadjusted sodium | 0.686 | 0.446–0.826 | 66 | 24 | 10 | 0.524 | <0.01 |
| Energy adjusted sodium | 0.541 | 0.191–0.739 | 50 | 36 | 14 | 0.316 | 0.026 |

ICC = intraclass correlation coefficient, 95% CI = 95% confidence interval

The quantification of sodium intake remains a challenging task. In addition to its great variability, sodium metabolic balance is sensitive to changes inherent to each individual [34]. Even for individuals who constantly consume salt (6, 9, and 12 g/day), urinary sodium excretion will not remain constant. The variation in excretion is a reflection of the biological rhythm; that is, salt intake does not necessarily reflect the corresponding excretion [34].

A single 24-h urine specimen is probably inadequate for validating the FFQ [10]. A single 24-h urine specimen, not with standing whether properly collected, cannot determine a 3-g difference in daily salt intake [34]. To assess for sodium intake, the 24-h urine collection should be repeated 2–7 times [10]. In the literature, urinary excretion tests to assess sodium intake varied from those using a single sample [1, 35] to six collected specimens [36]. Regardless of the number of repetitions of urinary excretion to assess for sodium intake, the correlations remained weak. The need for more urinary excretion repetitions can justify the difficulty in validating the instrument.

Other studies corroborate that the correlations between urinary excretion and 24hR/FR are stronger than those between FFQ survey and urinary excretion [23, 36]. Day et al. [36] used six 24-h urine specimens to measure the sodium, potassium, and nitrogen intakes. The FFQ survey and 7-day FR underestimated the sodium intake, and the correlation between urinary excretion and the FFQ survey (r = 0.13) was lower than that with the FR(r = 0.36). Some studies reported that the correlations between dietary intake assessment methods are stronger than those between methods using Bs (urinary sodium excretion) because the different dietary intake assessment tools have correlating and systematic errors, which are different and independent from the errors inherent to a B analysis [4, 35, 36].

Tangney et al. [32] also reported the underestimation of sodium consumption. The food items in the assessment tool did not identify the main sources of sodium. A dietary assessment tool can never be expected to capture 100% of the sodium intake, but it is expected to provide a reliable estimate [32]. The selection of foods that comprise the FFQ for habitual sodium intake must include foods and/or dishes usually consumed by the population, considering their contribution to sodium intake, as well as processed and ultra-processed foods, because it is believed that the higher sodium intake is attributed to the consumption of these food items [11]. According to Cooper et al. [37], one of the limitations in the development of the Sodium AnaLysis Tool (SALT) was the absence of ethnic dishes, which are typical preparations of the study population. This may have interfered with the responses of individuals who consumed food items other than those listed in the instrument. Inclusion of these food items remains a challenge as the use of an extensive list can make the interview tiring and diminish the quality of the interviewee's response, while the use of a short list makes it difficult to identify the variability of individual diets [38].

Dietary sodium has intrinsic peculiarities. The assessment of dietary sodium intake using an FFQ is limited by the fact that the list of food items with high sodium content is primarily composed of processed and ultra-processed foods, in which the *in natura* or minimally processed foods that comprise the main meals, lunch and dinner, of a Brazilians' diet are not represented. In addition to the sodium present in foods, it is necessary to consider the amount of salt as a culinary ingredient and the added salt at the table.

With regard to the study limitations, the 24hR was performed prior to the FFQ-FHS survey, which was in disagreement with the recommendations of some authors [14]. According to Willet [29], the time and effort undertaken by the research participants to complete the records and dietary recalls, either by direct weighing or by recording what was consumed, may influence the completion of subsequent FFQ. However, if the FFQ is administered prior to the reference method (24hR/FR), these records can be influenced by the FFQ responses and do not reflect the reality [29]. In addition, McLean et al. [10] recommended that in order to assess the

sodium intake, the 24-h urinary excretion test should be used as a reference tool; thus, the order of administration of FFQ and 24hR would not be important in this case.

In this study, both instruments were administered during in-person interviews, and not self-reported, using a photographic manual as an aid for completion of the 24hR and a specific album of photos showing the portion size of the FFQ-FHS items. As a positive aspect, the average dietary sodium intake was obtained by conducting a 24hR four times, which exceeds number of 24hRs performed in diverse validation studies (three repetitions of the 24hR or FR) [1, 9, 23]. The 24-h urinary excretion test was then repeated. Nonetheless, owing to the population profile, there was considerable sample loss, and a large portion of the urine specimens were collected on the weekend. This might have led to the misrepresentation of sodium that is usually consumed during the week. Because it is difficult to collect, the participants may not have collected all urine samples voided in 24 h, although they received oral and written instructions.

Currently, the existing FFQs are not valid for dietary sodium assessment because they show poor agreement. Thus, 24-h urine remains the gold standard for estimating the intake of this nutrient [10]. Perhaps the validation of an FFQ as the only tool for estimating sodium intake is not the best method. The singularities of this nutrient, with a high influence of intra-individual variability, made it difficult or even impossible to assess the actual intake [34].

The FFQ-FHS was demonstrated to be reproducible over a period of 6 months, with unadjusted correlation coefficients (0.686) and energy-adjusted ones (0.541) considered acceptable. Cooper et al. [37] also found a significant correlation between these two SALT applications (p = 0.001).

According to Cade et al. [14], the reproducibility of a short FFQ can be determined by considering the unadjusted data. Usually, the correlations of instruments re-administered within longer intervals, 6 months to 1 year apart, are lower than those of instruments re-administered within a shorter interval.

Hence, the FFQ-FHS should be used as a complement to other methods of assessing sodium intake. The combined use of the FFQ and 24hR enables a better detailing of the eating habits and may be especially useful for identifying the foods that are not consumed for many days and, therefore, might have been totally forgotten during the 24hR [29].

In conclusion, despite the administration of four 24hRs and the repeated 24-h urinary sodium excretion, the instrument was not considered valid for the assessment of sodium intake and cannot be used alone for this purpose; however, it may be convenient as a complement to the reference method. The FFQ-FHS was reproducible over a 6-month period.

## Supporting information

**S1 File.**
(ZIP)

## Acknowledgments

We thank all the participants who agreed to participate in this study. We also thank TMSV, GBC, RKFS, ACOB, Hypertension Nutrition Care Group (Grupo de Atendimento Nutricional em Hipertensão, GANuH), and Nutrition Laboratory (Laboratório de Nutrição, LABNUT) for assistance in collecting data and analyzing biological samples.

## Author Contributions

**Conceptualization:** Diana S. Souza, Liliane V. Pires, Diva A. S. Vieira, Analícia R. S. Freire, Kiriaque B. F. Barbosa.

**Data curation:** Diana S. Souza, Bianca I. Santos, Brenda M. Costa, Dalila M. Santos, Laryssa G. S. Aragão.

**Formal analysis:** Diana S. Souza, Liliane V. Pires.

**Funding acquisition:** Kiriaque B. F. Barbosa.

**Investigation:** Diana S. Souza.

**Methodology:** Liliane V. Pires, Diva A. S. Vieira, Analícia R. S. Freire.

**Project administration:** Liliane V. Pires, Analícia R. S. Freire, Kiriaque B. F. Barbosa.

**Supervision:** Liliane V. Pires, Diva A. S. Vieira, Analícia R. S. Freire, Kiriaque B. F. Barbosa.

**Writing – original draft:** Diana S. Souza, Bianca I. Santos, Brenda M. Costa, Dalila M. Santos, Laryssa G. S. Aragão.

**Writing – review & editing:** Liliane V. Pires, Diva A. S. Vieira, Analícia R. S. Freire, Kiriaque B. F. Barbosa.

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
