## [Decision Letter · Decision Letter 0]

25 May 2022

PONE-D-21-23343FOOD FREQUENCY QUESTIONNAIRE FOR FOODS HIGH IN SODIUM: VALIDATION WITH THE TRIADS METHODPLOS ONE

Dear Dr. Souza,

Thank you for submitting your manuscript to PLOS ONE. Firstly, we would like to apologize for the delay in processing your manuscript. It has been exceptionally difficult to secure reviewers to evaluate your study. We have now received one completed review, which is available below. The reviewer has raised significant scientific concerns about the study and has indicated that extensive revisions are required.

Please note that we have only been able to secure a single reviewer to assess your manuscript. We are issuing a decision on your manuscript at this point to prevent further delays in the evaluation of your manuscript. Please be aware that the editor who handles your revised manuscript might find it necessary to invite additional reviewers to assess this work once the revised manuscript is submitted. However, we will aim to proceed on the basis of this single review if possible.

After careful consideration, we feel that it has merit but does not fully meet PLOS ONE’s publication criteria as it currently stands. Therefore, we invite you to submit a revised version of the manuscript that addresses the points raised during the review process.

We look forward to receiving your revised manuscript.

Kind regards,

Miquel Vall-llosera Camps

Senior Editor

PLOS ONE

Journal Requirements:

4. Please provide additional details regarding participant consent. In the Methods section, please ensure that you have specified (1) whether consent was informed and (2) what type you obtained (for instance, written or verbal). If your study included minors, state whether you obtained consent from parents or guardians. If the need for consent was waived by the ethics committee, please include this information

6. Thank you for stating in your Funding Statement: "This study was partially financed by the Fundação de Apoio à Pesquisa e a Inovação Tecnológica do Estado de Sergipe (Fapitec/SE/Brasil) [Research and Technological Innovation Support Foundation in the State of Sergipe] and the Coordenação de Aperfeiçoamento de Pessoal de Nível Superior - Brasil (CAPES/Brasil) [Coordination for the Improvement of Higher Education Personnel], financing code 001, PROMOB, process no. 88881.157882/2017-01.

https://www.gov.br/capes/pt-br

https://fapitec.se.gov.br/

7. Thank you for stating the following in the Acknowledgments Section of your manuscript: "This study was partially financed by the Fundação de Apoio à Pesquisa e a Inovação Tecnológica do Estado de Sergipe (Fapitec/SE/Brasil) [Research and Technological Innovation Support Foundation in the State of Sergipe] and the Coordenação de Aperfeiçoamento de Pessoal de Nível Superior - Brasil (CAPES/Brasil) [Coordination for the Improvement of Higher Education Personnel], financing code 001, PROMOB, process no. 88881.157882/2017-01."

Please remove any funding-related text from the manuscript and let us know how you would like to update your Funding Statement. Currently, your Funding Statement reads as follows: "This study was partially financed by the Fundação de Apoio à Pesquisa e a Inovação Tecnológica do Estado de Sergipe (Fapitec/SE/Brasil) [Research and Technological Innovation Support Foundation in the State of Sergipe] and the Coordenação de Aperfeiçoamento de Pessoal de Nível Superior - Brasil (CAPES/Brasil) [Coordination for the Improvement of Higher Education Personnel], financing code 001, PROMOB, process no. 88881.157882/2017-01.

https://www.gov.br/capes/pt-br

https://fapitec.se.gov.br/

8. In your Data Availability statement, you have not specified where the minimal data set underlying the results described in your manuscript can be found. PLOS defines a study's minimal data set as the underlying data used to reach the conclusions drawn in the manuscript and any additional data required to replicate the reported study findings in their entirety. All PLOS journals require that the minimal data set be made fully available. For more information about our data policy, please see http://journals.plos.org/plosone/s/data-availability.

9. We note that you have stated that you will provide repository information for your data at acceptance. Should your manuscript be accepted for publication, we will hold it until you provide the relevant accession numbers or DOIs necessary to access your data. If you wish to make changes to your Data Availability statement, please describe these changes in your cover letter and we will update your Data Availability statement to reflect the information you provide.

10. Please ensure that you include a title page within your main document. You should list all authors and all affiliations as per our author instructions and clearly indicate the corresponding author.

11. Please amend either the abstract on the online submission form (via Edit Submission) or the abstract in the manuscript so that they are identical.

12.Please include captions for your Supporting Information files at the end of your manuscript, and update any in-text citations to match accordingly. Please see our Supporting Information guidelines for more information: http://journals.plos.org/plosone/s/supporting-information. 

Reviewers' comments:

Reviewer's Responses to Questions

**Comments to the Author**

1. Is the manuscript technically sound, and do the data support the conclusions?

Reviewer #1: Yes

2. Has the statistical analysis been performed appropriately and rigorously? 

Reviewer #1: Yes

3. Have the authors made all data underlying the findings in their manuscript fully available?

Reviewer #1: No

4. Is the manuscript presented in an intelligible fashion and written in standard English?

Reviewer #1: No

5. Review Comments to the Author

Reviewer #1: The article addresses an important field of nutritional research. Evaluating the relative validity of a dietary assessment method is of great importance when studying the potential impact of diet exposures on health outcomes. The authors constructed an FFQ for foods high in sodium and proposed a validation protocol over a 6-month period of dietary data collection complemented by four 24-hour recalls and two 24-hour urinary excretion for sodium analysis. The FFQ was reproductible but not valid. Despite the good statistical procedures adopted, the writing is not always fluid and sometimes repetitive. The description of the study protocol is not clear. In addition, it is not always clear what each test adds to the overall evaluation of level of validity, which could be made clearer in the method section. Another concern I have is related to the 'per se' construction of FFQ that was done previously with another sample; so, in the middle of the methodology, the authors describe basically another study (not mentioned before) and confuse the reader. Important decisions on how FFQ was built are not provided in the text.

Very important: The English needs considerable attention.

Introduction:

1. I suggest deleting the first two paragraphs and getting straight to the point. In the last paragraph, authors could justify why this study was done and, especially to investigate sodium in that region of Brazil. (why a specific FFQ is needed to that region)

Methods

2. I believe that the authors could benefit from checking the manuscript construction of previous FFQ validation studies. As mentioned above, the writing is not fluid and sometimes repetitive. First, provide an overall description of the (i) previous protocol to design the FFQ; (ii) the 6-month protocol to the FFQ validation: FFQ, rec24h, urine collection (in a 6-month period, when each instrument was applied/data was collected, providing the interval period among them).

3. In the sentence: “Each volunteer answered three 24hR, with a minimum and maximum interval of 15 and 45 days, respectively, and one of 24hR covered one weekend day.” How was the 15-45-day interval decided? Does it depend on the availability of the participant?

4. Do FFQ and Rec24h were paper-based methods? Were both a face-to-face interview?

5. Very important: do not describe “researchers” when referring to those people that collected the data in the field work; in this case, I would recommend to describe they as “research assistants” or “assistants”. It is better to restrict the term 'researchers' to those responsible for the study protocol and who wrote the manuscript.

6. How many research assistants and nutritionists conducted the field work? Was the team composed of undergrad and grad students? Those who applied the dietary instruments had any background in nutrition & dietetics?

7. Examples of information that seems repetitive in the text:

(a) All the subjects who agreed to participate in the study received information on the study and signed the Free and Informed Consent Form.

(b) The participants signed the Free and Informed Consent Form, allowing their voluntary participation,

(a) All researchers had been trained and supervised by nutritionists.

(b) Nutritionists trained and supervised the anthropometric measurements to ensure higher precision and accuracy of the data.

8. Rewrite the sentence: To assess daily sodium intake, to the 24hR estimate was added the amount of sodium from added salt.

9. Briefly describe the data collected from the socioeconomic questionnaire applied. Age, sex, occupation, etc.

10. Provide more details on the multiple pass method applied in the current protocol.

11. Validation paragraph: It does not add additional information. In the way described, this paragraph can be deleted.

12. How was the information of the dietary intake of rec24h and FFQ entered in NutWin? Food intake data entered one-by-one? who and how many people entered these data? There was any quality control in the data entered? how portion sizes were defined? The same questions is applied to FFQ.

13. Describe more about the MSM. What this method proposes and how?

Results:

14. It calls my attention that the sample studied had higher education but low income. Who were they? Undergrad students or staff personnel? It is important to describe it in method section.

15. Table 1. Replace 'gender' by 'sex'. sex = biologic nature of being female or male. gender = social role construction of being male or female. If you are analyzing urinary excretion, it is more relevant that you provide the information on biology. Also, replace 'nutrition status' by 'body mass index (kg / m2)'.

16. Replace “CI95%” by 95% CI

17. I think authors can think about a better presentation on tables.

6. PLOS authors have the option to publish the peer review history of their article (what does this mean?). If published, this will include your full peer review and any attached files.

Reviewer #1: No

---

## [Author Response · Author response to Decision Letter 0]

30 Aug 2022

Response to Reviewers

Journal Requirements:

Response: The suggested changes have been made.

Response: The manuscript has been edited by Editage to ensure language and grammatical accuracy.

Response: As requested, a copy in the original language and English of the constructed FFQ has been attached.

4. Please provide additional details regarding participant consent. In the Methods section, please ensure that you have specified (1) whether consent was informed and (2) what type you obtained (for instance, written or verbal). If your study included minors, state whether you obtained consent from parents or guardians. If the need for consent was waived by the ethics committee, please include this information.

Response: The suggested changes have been made.

Response: This study was partially financed by the following:

1. Scholarship awarded by the Fundação de Apoio à Pesquisa e a Inovação Tecnológica do Estado de Sergipe (Fapitec/SE/Brasil) [Research and Technological Innovation Support Foundation in the State of Sergipe] 

https://fapitec.se.gov.br/

2. Financial resources granted by the Coordenação de Aperfeiçoamento de Pessoal de Nível Superior - Brasil (CAPES/Brasil) [Coordination for the Improvement of Higher Education Personnel], financing code 001, PROMOB, process no. 88881.157882/2017-01 

https://www.gov.br/capes/pt-br

The funders had no role in the study design, data collection and analysis, decision to publish, or manuscript preparation. No additional external funding was received for the study.

Response: The suggested changes have been made.

Please remove any funding-related text from the manuscript and let us know how you would like to update your Funding Statement. Currently, your Funding Statement reads as follows: "This study was partially financed by the Fundação de Apoio à Pesquisa e a Inovação Tecnológica do Estado de Sergipe (Fapitec/SE/Brasil) [Research and Technological Innovation Support Foundation in the State of Sergipe] and the Coordenação de Aperfeiçoamento de Pessoal de Nível Superior - Brasil (CAPES/Brasil) [Coordination for the Improvement of Higher Education Personnel], financing code 001, PROMOB, process no. 88881.157882/2017-01.

https://www.gov.br/capes/pt-br

https://fapitec.se.gov.br/

Response: The suggested changes have been made.

Acknowledgements

We thank all the participants who agreed to participate in this study. We also thank TMSV, GBC, RKFS, ACOB, Hypertension Nutrition Care Group (Grupo de Atendimento Nutricional em Hipertensão, GANuH), and Nutrition Laboratory (Laboratório de Nutrição, LABNUT) for assistance in collecting data and analyzing biological samples.

8. In your Data Availability statement, you have not specified where the minimal data set underlying the results described in your manuscript can be found. PLOS defines a study's minimal data set as the underlying data used to reach the conclusions drawn in the manuscript and any additional data required to replicate the reported study findings in their entirety. All PLOS journals require that the minimal data set be made fully available. For more information about our data policy, please see http://journals.plos.org/plosone/s/data-availability.

Response: The database containing the collected data will be attached.

9. We note that you have stated that you will provide repository information for your data at acceptance. Should your manuscript be accepted for publication, we will hold it until you provide the relevant accession numbers or DOIs necessary to access your data. If you wish to make changes to your Data Availability statement, please describe these changes in your cover letter and we will update your Data Availability statement to reflect the information you provide.

Response: The suggested changes have been made.

10. Please ensure that you include a title page within your main document. You should list all authors and all affiliations as per our author instructions and clearly indicate the corresponding author.

Response: The suggested changes have been made.

11. Please amend either the abstract on the online submission form (via Edit Submission) or the abstract in the manuscript so that they are identical.

Response: The suggested changes have been made.

12.Please include captions for your Supporting Information files at the end of your manuscript, and update any in-text citations to match accordingly. Please see our Supporting Information guidelines for more information: http://journals.plos.org/plosone/s/supporting-information.

Response: The suggested changes have been made.

Reviewers' comments:

Reviewer #1: The article addresses an important field of nutritional research. Evaluating the relative validity of a dietary assessment method is of great importance when studying the potential impact of diet exposures on health outcomes. The authors constructed an FFQ for foods high in sodium and proposed a validation protocol over a 6-month period of dietary data collection complemented by four 24-hour recalls and two 24-hour urinary excretion for sodium analysis. The FFQ was reproductible but not valid. Despite the good statistical procedures adopted, the writing is not always fluid and sometimes repetitive. The description of the study protocol is not clear. In addition, it is not always clear what each test adds to the overall evaluation of level of validity, which could be made clearer in the method section. Another concern I have is related to the 'per se' construction of FFQ that was done previously with another sample; so, in the middle of the methodology, the authors describe basically another study (not mentioned before) and confuse the reader. Important decisions on how FFQ was built are not provided in the text.

Very important: The English needs considerable attention.

Response: For validation, the triad method was used by triangulation between the FFQ-FHS, 24hR, and biomarker. For reproducibility, the intraclass coefficient of correlation (ICC) and Kappa coefficient were used; Bland–Altman plots were constructed to graphically demonstrate the level of agreement between the FFQ-FHS, 24hR, and biomarker.

More information about the constructed FFQ-FHS has been added to the methodology.

Introduction:

1. I suggest deleting the first two paragraphs and getting straight to the point. In the last paragraph, authors could justify why this study was done and, especially to investigate sodium in that region of Brazil. (why a specific FFQ is needed to that region)

Response: The suggested changes have been made. The two paragraphs have been deleted, and the justification has been inserted in the last paragraph.

In Brazil, independent of the geographical region, the sodium intake of the population is high [11,12] and corresponds to approximately twice the intake recommended by the World Health Organization (WHO) (2 g/day) [13]. Data from the 2008 to 2009 Household Budget Survey show that the contribution of processed and ultra-processed foods to the average sodium intake increased from that observed in the first edition of the survey (2002–2003) [11]. Despite this, studies seeking the validation of a tool to investigate sodium intake in the Brazilian population, especially in the northeast region, remain scarce.

Methods

2. I believe that the authors could benefit from checking the manuscript construction of previous FFQ validation studies. As mentioned above, the writing is not fluid and sometimes repetitive. First, provide an overall description of the (i) previous protocol to design the FFQ; (ii) the 6-month protocol to the FFQ validation: FFQ, rec24h, urine collection (in a 6-month period, when each instrument was applied/data was collected, providing the interval period among them).

Response: The suggested changes have been made.

3. In the sentence: “Each volunteer answered three 24hR, with a minimum and maximum interval of 15 and 45 days, respectively, and one of 24hR covered one weekend day.” How was the 15-45-day interval decided? Does it depend on the availability of the participant?

Response: Each volunteer answered three 24hR questions, with a minimum and maximum interval of 15 and 45 days, respectively, considering the participants' availability, including one day during the weekend.

4. Do FFQ and Rec24h were paper-based methods? Were both a face-to-face interview?

Response: Yes, all interviews regarding the application of the R24h and FFQ instruments were conducted face-to-face.

5. Very important: do not describe “researchers” when referring to those people that collected the data in the field work; in this case, I would recommend to describe they as “research assistants” or “assistants”. It is better to restrict the term 'researchers' to those responsible for the study protocol and who wrote the manuscript.

Response: The suggested changes have been made.

6. How many research assistants and nutritionists conducted the field work? Was the team composed of undergrad and grad students? Those who applied the dietary instruments had any background in nutrition & dietetics?

Response: All four assistants were nutrition graduate students who received training and supervision from the nutritionists.

7. Examples of information that seems repetitive in the text:

(a) All the subjects who agreed to participate in the study received information on the study and signed the Free and Informed Consent Form.

(b) The participants signed the Free and Informed Consent Form, allowing their voluntary participation,

(a) All researchers had been trained and supervised by nutritionists.

(b) Nutritionists trained and supervised the anthropometric measurements to ensure higher precision and accuracy of the data.

Response: The suggested changes have been made.

8. Rewrite the sentence: To assess daily sodium intake, to the 24hR estimate was added the amount of sodium from added salt.

Response: The suggested changes have been made.

9. Briefly describe the data collected from the socioeconomic questionnaire applied. Age, sex, occupation, etc.

Response: A structured questionnaire was used to collect sociodemographic information (sex, age, household per capita income, and educational level).

10. Provide more details on the multiple pass method applied in the current protocol.

Response: The MPM is an approach technique for applying 24hR, aimed at increasing the accuracy of information. The MPM was proposed in five steps.

1 Quick list

2 Forgotten foods list

3 Time and occasion

4 Detail and review

5 Final review

11. Validation paragraph: It does not add additional information. In the way described, this paragraph can be deleted.

Response: The suggested changes have been made.

12. How was the information of the dietary intake of rec24h and FFQ entered in NutWin? Food intake data entered one-by-one? who and how many people entered these data? There was any quality control in the data entered? how portion sizes were defined? The same questions is applied to FFQ.

Response: To facilitate the completion of all forms, a photographic manual of food quantification with a preset portion size was used.

The standard portion size of each food or meal was defined according to the Brazilian Institute of Geography and Statistics (Instituto Brasileiro de Geografia e Estatística, IBGE) database of reference measurements and portion sizes of foods and dishes consumed in Brazil.

The food nutrient composition was analyzed using NutWin® software (version 1.5.2.51). All items listed in the applied 24hR and FFQ-FHS were standardized and typed individually in duplicate.

13. Describe more about the MSM. What this method proposes and how?

Response: More information about the MSM has been included in the statistical analysis section.

Results

14. It calls my attention that the sample studied had higher education but low income. Who were they? Undergrad students or staff personnel? It is important to describe it in method section.

Response: The suggested changes have been made.

15. Table 1. Replace 'gender' by 'sex'. sex = biologic nature of being female or male. gender = social role construction of being male or female. If you are analyzing urinary excretion, it is more relevant that you provide the information on biology. Also, replace 'nutrition status' by 'body mass index (kg / m2)'.

Response: The suggested changes have been made.

16. Replace “CI95%” by 95% CI

Response: The suggested changes have been made.

17. I think authors can think about a better presentation on tables.

Response: The suggested changes have been made.

---

## [Decision Letter · Decision Letter 1]

28 Oct 2022

PONE-D-21-23343R1Food frequency questionnaire for foods high in sodium: validation with the triads methodPLOS ONE

Dear Dr. Souza,

Thank you for submitting your manuscript to PLOS ONE. After careful consideration, we feel that it has merit but does not fully meet PLOS ONE’s publication criteria as it currently stands. Therefore, we invite you to submit a revised version of the manuscript that addresses the points raised during the review process.

The reviewers highlighed the great quality of the new version of the manuscript. However, one of them highlighted some methodological information are required and are not described in the manuscript. All comments adressed to authors are describe below.

We look forward to receiving your revised manuscript.

Kind regards,

Ana Elisa Madalena Rinaldi, Ph.D.

Academic Editor

PLOS ONE

Journal Requirements:

Reviewers' comments:

Reviewer's Responses to Questions

**Comments to the Author**

1. If the authors have adequately addressed your comments raised in a previous round of review and you feel that this manuscript is now acceptable for publication, you may indicate that here to bypass the “Comments to the Author” section, enter your conflict of interest statement in the “Confidential to Editor” section, and submit your "Accept" recommendation.

Reviewer #1: All comments have been addressed

Reviewer #2: All comments have been addressed

2. Is the manuscript technically sound, and do the data support the conclusions?

Reviewer #1: Yes

Reviewer #2: Yes

3. Has the statistical analysis been performed appropriately and rigorously? 

Reviewer #1: Yes

Reviewer #2: Yes

4. Have the authors made all data underlying the findings in their manuscript fully available?

Reviewer #1: No

Reviewer #2: Yes

5. Is the manuscript presented in an intelligible fashion and written in standard English?

Reviewer #1: No

Reviewer #2: Yes

6. Review Comments to the Author

Reviewer #1: The current version of the manuscript had a great improvement from previous one. However, there are still some gaps to be described in the Methods section, considering that this is a methodological study, and it may be used as a reference for future studies in Brazil.

Methods:

Sodium biomarkers:

1. Please describe the instructions provided for participants of how to collect 24-h urine samples. Also, provide a reference for it.

2. The reference methods applied to verify the completeness of urine samples are not clear. First, is there a reference of less than 400 ml/day for completeness? Second, the study of Charlton KE et al 2008 (21) used multiple parameters, as volume of urine less than 500 ml/day and creatinine 0.2–0.3 mmol kg/day. These parameters and references are very important considering that urine provided the biomarker. If author be more rigid to accept a sample as valid (e.g., equal or more 500 ml/day + creatine output >0.2 mmol kg/day), how many samples would be valid in both methods? Did you test whether concordance with dietary method had improved?

3. Sociodemographic data. Please indicate the currency conversion of Brazilian currency to USD$ at the time of the study.

4. Anthropometric measurements: indicate the equipment (registered brand) of each anthropometric tool (electronic scale, stadiometer, and tape)

5. Food composition analysis of FFQ: Has NutWin a specific interface for FFQ calculation? Or were all food items converted to daily portion size in order to be entered in this software? Please, describe in methods.

6. Statistics: Which method was applied to calculate energy adjustment?

7. Line 200: Correct typing mistake “With Rregard to nutritional 200 status…”

8. Table 1: present variables summarized as mean first, then after the categorical ones; Use only 1 decimal to summarize age

9. Results: Line 209-211: Adequacy in urine sample collection: First, there is no need to repeat the parameter in parenthesis already described in methods; second, indicate how many were adequate according to volume and how many according to creatinine value. Also, as described above, test whether being more rigid in urine assessments could improve the final results with dietary methods.

Reviewer #2: Thank you for addressing all the comments from the reviewers. The suggestions were incorporated to the manuscript.

7. PLOS authors have the option to publish the peer review history of their article (what does this mean?). If published, this will include your full peer review and any attached files.

Reviewer #1: **Yes: **Juliana dos Santos Vaz

Reviewer #2: No

---

## [Author Response · Author response to Decision Letter 1]

24 Feb 2023

Response to Reviewers

Methods:

Sodium biomarkers:

1. Please describe the instructions provided for participants of how to collect 24-h urine samples. Also, provide a reference for it.

Response: 

A 24-hour urine collection was adopted as an instrument to assess creatinine clearance and estimate Na+ electrolyte intake. Participants were instructed regarding urine collection and received a 2 L plastic bottle. Furthermore, the participants received a questionnaire containing written information about the correct urine collection procedure to be completed and delivered on the scheduled day. The following collection instructions were standardized:

- At the beginning of the collection, the participant discarded all urine stored in the bladder in the toilet. From then on, the participant stored all urine, including night and bath urine, in the bottle until the next day.

- The collected urine wasstored in a refrigerator, both during and after collection.

Reference: Sociedade Brasileira de Patologia Clínica/Medicina Laboratorial(SBPC/ML). Recomendações da Sociedade Brasileira de Patologia Clínica/Medicina Laboratorial (SBPC/ML): coleta e preparo da amostra biológica. Barueri, SP: Manole: Minha Editora. 2014: 487p.

2. The reference methods applied to verify the completeness of urine samples are not clear. First, is there a reference of less than 400 ml/day for completeness? Second, the study of Charlton KE et al 2008 (21) used multiple parameters, as volume of urine less than 500 ml/day and creatinine 0.2–0.3 mmol kg/day. These parameters and references are very important considering that urine provided the biomarker. If author be more rigid to accept a sample as valid (e.g., equal or more 500 ml/day + creatine output >0.2 mmol kg/day), how many samples would be valid in both methods? Did you test whether concordance with dietary method had improved?

Response:

For the metabolites to be excreted, a minimum urine volume of 400mL is required, according to Nemeret al. (2010). For this research, urine samples lessthan 400mL and/or those whose collection period exceeded the determined 24 h were excluded. Creatinine excretion was also analyzed as an indicator of adequate urine collection. Urine samples with creatinine levels<0.2 mmol/kg/day were excluded. The results were converted to mg/kg/day (CHARLTON, et al., 2008).

When performing urinary creatinine analysis, only one participant met the exclusion criterion of creatinine <0.2 mmol/kg/day, obtaining a result of 0.171 mmol/kg/day in the second urine sample collected.

3. Sociodemographic data. Please indicate the currency conversion of Brazilian currency to USD$ at the time of the study.

Response:

The suggested changes have been made.

4. Anthropometric measurements: indicate the equipment (registered brand) of each anthropometric tool (electronic scale, stadiometer, and tape)

Response:

Digital platform scale, with 100g precision (LIDER®, São Paulo, Brazil)

Portable Stadiometer, with 0.1 cm resolution (Alturaexata®, Minas Gerais, Brazil)

5. Food composition analysis of FFQ: Has NutWin a specific interface for FFQ calculation? Or were all food items converted to daily portion size in order to be entered in this software? Please, describe in methods.

Response:

For the analysis of the nutritional composition of the food frequency questionnaire (FFQ), all items were converted into individual consumption portions and subsequently tabulated using the software.

6. Statistics: Which method was applied to calculate energy adjustment?

Response:

Sodium intake was adjusted using the residual method to control for confounding factors and remove external oscillations, according to Willetet al. (1997). Simple linear regression analysis was performed, considering energy as an independent variable and sodium as a dependent variable.

7. Line 200: Correct typing mistake “With Rregard to nutritional 200 status…”

Response:

The suggested changes have been made.

8. Table 1: present variables summarized as mean first, then after the categorical ones; Use only 1 decimal to summarize age

Response:

The suggested changes have been made.

9. Results: Line 209-211: Adequacy in urine sample collection: First, there is no need to repeat the parameter in parenthesis already described in methods; second, indicate how many were adequate according to volume and how many according to creatinine value. Also, as described above, test whether being more rigid in urine assessments could improve the final results with dietary methods.

Response:

Considering a urinary volume of ≥ 500mL, 43 and 36 urine samples from the first collection and second collections, respectivelywere considered adequate. By joining two exclusion criteria (e.g., equal or more 500 mL/day + creatine output >0.2 mmol kg/day), 43 urine samples from the first collection and 32 samples from the second collection were considered adequate.

New statistical tests were performed on the 32 urine samples. The table below presents the results.

Table 1. Spearman’s correlation and validity coefficients for unadjusted and energy-adjusted daily sodium intake in adults in the academic community. Aracaju, SE, 2018 (n=50) 

Correlation coefficient Validity coefficient

 Unadjusted (p) Adjusted (p)* Adjusted*

rFFQR 0.386 (<0.01) 0.129 (0.387) ρFFQAI 0.26

rRB† −0.026 (0.882) 0.172 (0.316) ρRAI† 0.85

rFFQB† 0.014 (0.936) 0.052 (0.763) ρBAI† 0.20

rFFQR, correlation between the food frequency questionnaire survey and the 24-h dietary recall

rRB, correlation between the 24-h dietary recall and biomarker assessment

rFFQB, correlation between the food frequency questionnaire survey and biomarker assessment

ρFFQAI, validity coefficient of the food frequency questionnaire survey 

ρRAI, validity coefficient of the 24-h dietary recall

ρBAI, biomarker validity coefficient

ρFFQAI=√([rFFQR x rFFQB])/rRB

ρRAI=√([rFFQR x rRB])/rFFQB

ρBAI=√([rRB x rFFQB])/rFFQR

*Daily energy-adjusted sodium intake

†n,36 using the average urinary sodium excretion

Table 2. Spearman’s correlation and validity coefficients for unadjusted and energy-adjusted daily sodium intake in adults in the academic community. Aracaju, SE, 2018 (n=32) 

Correlation coefficient Validity coefficient

 Unadjusted (p) Adjusted (p)* Adjusted*

rFFQR 0.242 (0.182) 0.244 (0.178) ρFFQAI 0.23

rRB† 0.139 (0.449) 0.232 (0.201) ρRAI 1.06

rFFQB† 0.225 (0.215) 0.050 (0.786) ρBAI 0.22

The repetition of statistical tests demonstrates that even after considering new inclusion parameters for adequate urine samples (e.g., equal or more 500 ml/day + creatine output >0.2 mmol kg/day), the correlations and validity coefficients remained insufficient for the FFQAI validation.

---

## [Decision Letter · Decision Letter 2]

19 Apr 2023

PONE-D-21-23343R2Questionário de frequência alimentar para alimentos ricos em sódio: validação com o método das tríadesPLOS ONE

Dear Dr. Souza,

Thank you for submitting your manuscript to PLOS ONE. After careful consideration, we feel that it has merit but does not fully meet PLOS ONE’s publication criteria as it currently stands. Therefore, we invite you to submit a revised version of the manuscript that addresses the points raised during the review process.

 All the previous queries have been addressed, there are only a few minor details that need to be taken care of. Once these are done, I consider that the ms can be accepted.

We look forward to receiving your revised manuscript.

Kind regards,

Nicoletta Righini, PhD

Academic Editor

PLOS ONE

Journal Requirements:

Reviewers' comments:

Reviewer's Responses to Questions

**Comments to the Author**

1. If the authors have adequately addressed your comments raised in a previous round of review and you feel that this manuscript is now acceptable for publication, you may indicate that here to bypass the “Comments to the Author” section, enter your conflict of interest statement in the “Confidential to Editor” section, and submit your "Accept" recommendation.

Reviewer #1: All comments have been addressed

2. Is the manuscript technically sound, and do the data support the conclusions?

Reviewer #1: Yes

3. Has the statistical analysis been performed appropriately and rigorously? 

Reviewer #1: Yes

4. Have the authors made all data underlying the findings in their manuscript fully available?

Reviewer #1: Yes

5. Is the manuscript presented in an intelligible fashion and written in standard English?

Reviewer #1: Yes

6. Review Comments to the Author

**Reviewer #1:** The authors addressed all comments raised in a previous round of review. The current version of the manuscript now reports all methodological aspects required in a validation study.

I just request some minor corrections in the text, and 1-2 other adjustments in tables and text.

Please add spacing in the following passages:

Line 42: “However, Bshave limitations…” = B have

Line 43: “…Blevels…” = B levels

Line 47: “… 24hR or FR, and Bs,…” = B (all other instruments described in this sentence are in singular version)

Line 58: “… to2009…”

Line 62: “… remainscarce”

Line 75: “… 2018.Fifty…”

Line 85: “… period of 6months..” = 6 months

Line 89: Fig1 Data collectionprocess

Line 94: “…24hR and FFQ-FHS survey,” = there is no need the term “survey”

Line 96: “… was completed after the 24hRviaface-to-face…”

Line 123: “After 3months…” = 3 months

Line 124: “…24hRsusing…”

Line 140: “[10]to determine”

Line 164: “…precision of 1millimeter” = 1 millimeter

Line 325: “… notwithstanding…”

Line 370: “… duringin-person interviews,…”

Line 386: “… over a period of 6months” = 6 months

Line 77: Please, reedit the sentence: “We excluded pregnant and lactating women…"

Other minor corrections:

Please review in all Tables and Bland and Altman figure the following: Clear state in the tables when the “adjusted” means “energy adjusted”, otherwise it means traditional statistical adjustments for covariates.

Line 296: “… which eventually lost significance after adjustment.” Does it mean energy adjustment? Please specify in the text.

7. PLOS authors have the option to publish the peer review history of their article (what does this mean?). If published, this will include your full peer review and any attached files.

Reviewer #1: **Yes: **Juliana Vaz

---

## [Author Response · Author response to Decision Letter 2]

19 Jun 2023

I just request some minor corrections in the text, and 1-2 other adjustments in tables and text.

Please add spacing in the following passages:

Line 42: “However, Bshave limitations…” = B have

Line 43: “…Blevels…” = B levels

Line 47: “… 24hR or FR, and Bs,…” = B (all other instruments described in this sentence are in singular version)

Line 58: “… to2009…”

Line 62: “… remainscarce”

Line 75: “… 2018.Fifty…”

Line 85: “… period of 6months..” = 6 months

Line 89: Fig1 Data collectionprocess

Line 94: “…24hR and FFQ-FHS survey,” = there is no need the term “survey”

Line 96: “… was completed after the 24hRviaface-to-face…”

Line 123: “After 3months…” = 3 months

Line 124: “…24hRsusing…”

Line 140: “[10]to determine”

Line 164: “…precision of 1millimeter” = 1 millimeter

Line 325: “… notwithstanding…”

Line 370: “… duringin-person interviews,…”

Line 386: “… over a period of 6months” = 6 months

Line 77: Please, reedit the sentence: “We excluded pregnant and lactating women…"

Response: 

The requested corrections have been made.

Other minor corrections:

Please review in all Tables and Bland and Altman figure the following: Clear state in the tables when the “adjusted” means “energy adjusted”, otherwise it means traditional statistical adjustments for covariates.

Response: 

The suggested changes have been made.

Line 296: “… which eventually lost significance after adjustment.” Does it mean energy adjustment? Please specify in the text.

Response: 

Yes, does it mean energy adjusment. The suggested changes have been made.

---

## [Editor Report · Decision Letter 3]

20 Jun 2023

Questionário de frequência alimentar para alimentos ricos em sódio: validação com o método das tríades

PONE-D-21-23343R3

Dear Dr. Souza,

We’re pleased to inform you that your manuscript has been judged scientifically suitable for publication and will be formally accepted for publication once it meets all outstanding technical requirements.

Kind regards,

Nicoletta Righini, PhD

Academic Editor

PLOS ONE

Additional Editor Comments (optional):

All the comments were addressed by the authors and the minor corrections were performed.
---

## [Editor Report · Acceptance letter]

23 Jun 2023

PONE-D-21-23343R3 

Food frequency questionnaire for foods high in sodium: validation with the triads method 

Dear Dr. Souza:

I'm pleased to inform you that your manuscript has been deemed suitable for publication in PLOS ONE. Congratulations! Your manuscript is now with our production department. 

Kind regards, 

on behalf of

Dr. Nicoletta Righini 

Academic Editor

PLOS ONE